# Economic evaluation of *Wolbachia* deployment in Colombia: A modeling study

Donald S. Shepard[1]*, Samantha R. Lee[1], Yara A. Halasa-Rappel[1], Carlos Willian Rincon Perez[2], Arturo Harker Roa[2]

1 Heller School for Social Policy and Management, Brandeis University, Waltham, Massachusetts, United States of America, 2 School of Government, University of Los Andes, Bogotá, Colombia

* shepard@brandeis.edu

## Abstract

### Background and aims

*Wolbachia* are bacteria that inhibit dengue virus replication within the mosquito. A cluster-randomized trial in Indonesia found *Wolbachia* reduced virologically-confirmed dengue cases by 77.1%. Previous models predicted *Wolbachia* to be highly cost-effective in Indonesia, Vietnam, and Brazil. To inform decisions about future extensions in Colombia, we performed economic evaluations of potential *Wolbachia* deployments in 11 target cities.

### Methods

We assembled the numbers and distribution by severity of reported dengue cases from Colombia's national disease surveillance system and the health service provision registry (RIPS). An epidemiological panel of three experts estimated the shares of dengue that were non-medical, under-reported, or misreported as another disease. We determined costs (in 2020 US dollars at market prices) of treating dengue illness from the benchmark insurance tariff and RIPS data on treatment services per symptomatic dengue case. Our central estimates projected 10 years of efficacy and focused on Cali, the target city with the highest number of dengue cases.

### Results

For Cali, we estimated a net health-sector savings of US$4.95 per person and averting 369 disability-adjusted life years (DALYs) per 100,000 population. From a societal perspective, at 10 years *Wolbachia* deployment is expected to have highly favorable benefit-cost ratios, with benefits per dollar invested of US$5.50 in Cali and US$4.68 over all target cities.

**Data availability statement:** The authors do not have permission to distribute the external data bases used here but describe the data and contact information (generally from Colombian government agencies) below. Numbers in brackets correspond to reference numbers in our article. Through the information in the methods and the data in the manuscript, supplement, and the external sources provided, after appropriate registration interested researchers could replicate the findings of this study. Reported dengue cases were obtained from Sistema Nacional de Vigilancia en Salud Pública (SIVIGILA) [the National Public Health Surveillance System. Anonymous data are publicly available through a portal maintained by the Colombia's National Institute of Health. [28] Population data counts are publicly available here (https://www.dane.gov.co/index.php/estadisticas-por-tema/demografia-y-poblacion/proyecciones-de-poblacion). Standard insurance tariffs data are available from Seguro Obligatorio para Accidentes de Tránsito [Compulsory Insurance for Traffic Accidents] (SOAT), including payments for medical services serving as the reference prices used by Colombian insurers. Key data items are in Supporting Information S5 Table. The authors obtained access to this complete list of prices through a registration process. Access information is available here (https://www.minsalud.gov.co/proteccionsocial/Paginas/rips.aspx). Health care service utilization data are available from Registro Individual de Prestación de Servicios de Salud [Individual Registry of Provision of Health Services] (RIPS). This huge online data base contains every individual formal sector service of every Colombian resident. Suficiencia [Sufficiency] combines utilization and unit costs to calculate Colombia's capitation payments. Descriptive information is in the Methods at Disease Burden of Dengue [30]. The authors obtained online anonymous access to these databases through a registration process. Access information is available here (https://www.minsalud.gov.co/proteccionsocial/Paginas/rips.aspx). Colombia's health expenditure data was obtained from a report on the structure of health expenditures which is publicly available [32]. The authors extracted the necessary information from this report. Health insurance affiliation figure data is available from an interactive database which is

## Conclusions

Over 10 years, *Wolbachia* is highly beneficial on economic grounds, and almost universally cost saving. The *Wolbachia* program's economic benefits exceeded its costs in all 11 cities. The program's savings in healthcare costs alone would more than offset deployment costs nationally and in 9 of 11 target cities. *Wolbachia* is likely to be the most cost-effective or cost-saving dengue control option in municipalities with both high incidence of dengue and high population density, whereas areas with high dengue incidence but low population density should consider vaccination.

## Introduction

Dengue, responsible for dengue fever and dengue hemorrhagic fever, is the most widespread vector-borne virus in the southern hemisphere [1]. Colombia has experienced recent dengue epidemics in 2010, 2013, and 2019 [2].

*Wolbachia* are common bacteria that naturally infect fruit flies and many other insects. Researchers at the World Mosquito Program (WMP) discovered that they could infect *Aedes aegypti* mosquitoes with these bacteria [3] and that dengue, chikungunya and Zika viruses are then less able to replicate within the mosquitoes, thereby inhibiting the transmission of these mosquito-borne infections [4]. To use this method for disease control, governments, communities, and international organizations (e.g., the WMP) partner to grow mosquitoes infected with *Wolbachia* in insectaries and then deploy eggs or release adult mosquitoes to establish the bacteria in the local mosquito population. *Wolbachia*-infected mosquitoes transmit the bacteria through their eggs to the next generation. This approach is termed the "replacement" strategy, as it tends to replace wild mosquitoes by *Wolbachia*-infected ones. Thus, the establishment of *Wolbachia* becomes a sustainable and often long-term control mechanism at that site. The replacement approach was first applied near Cairns, Australia, using *Ae. aegypti* infected with the wMel strain of *Wolbachia*. Over a decade after initial deployment, mosquitoes there remain infected with the bacteria, supporting the long-term viability of the approach [5]. The replacement approach is being applied in countries in the Americas, Asia, and Oceania [6].

Under a different approach, the *Wolbachia* suppression strategy, Singapore releases only male *Wolbachia* infected mosquitoes (wAlbB strain) [3]. When these mosquitoes mate with wild mosquitoes, the eggs do not hatch, thereby reducing the number of potentially disease-carrying insects. While experience to date has found this approach efficacious, the need for annual releases makes the suppression approach more costly but potentially economically viable in this high-income country [7]. The remainder of this paper considers only the replacement approach, which is potentially suitable for low- and middle-income countries.

A landmark cluster-randomized trial in Yogyakarta, Indonesia found that the *Wolbachia* (wMel strain) replacement strategy reduced all virologically-confirmed symptomatic dengue cases by 77.1% and hospitalized cases by 86.2% under the original protocol analysis [8]. A reanalysis that corrected for the attenuation due to border

publicly available [33]. Worldometer Colombia population data used in this study are also publicly available [36].

**Funding:** This study was funded by the Wellcome Trust, a registered charity in England and Wales, under a grant (224459/Z/21/Z) to the World Mosquito Program, Monash University (Clayton, VIC, Australia) with a subaward to Brandeis University, USA. Costing methods were also funded in part by the Bill & Melinda Gates Foundation under a grant (OPP1187889) to Brandeis University. For the purpose of open access, the author has applied a CC BY public copyright license to any Author Accepted Manuscript version arising from this submission. Role of the Funders: The Funders had no role in review nor the decision to submit. The direct sponsor (WMP) had the right to review but authorized submission with no required changes.

**Competing interests:** Competing Interests: All authors received funding from the Wellcome Trust, a registered charity in England Wales, under a grant (224459/Z/21/Z) to the World Mosquito Program (WMP), Monash University (Clayton, VIC, Australia) with a subaward to Brandeis University, USA. Donald S. Shepard and Yara A Halasa-Rappel also received funding from the Bill & Melinda Gates Foundation under a grant (OPP1187889) to Brandeis University. This does not alter our adherence to PLOS ONE policies on sharing data and materials. Donald S. Shepard has received financial support from Abbott, Inc, Sanofi, and Takeda Vaccines, Inc. in the past 36 months unrelated to the present study. All other authors declare no other competing interests.

crossing by humans and mosquitoes raised the estimated efficacy against dengue cases to 82.7% [9]. Following the successful completion of the cluster randomized trial in Yogyakarta, *Wolbachia* were deployed in its previous control clusters. *Wolbachia* proved 76% efficacious even in areas with only partial (60%-80%) *Wolbachia* coverage [10].

A quasi-experimental study from Niterói, Brazil found that wMel *Wolbachia* reduced the incidence of dengue by 69%, of chikungunya by 56%, and of Zika by 37% [11]. Research in Rio de Janeiro has shown that the technique is generally robust. Even in neighborhoods where *Wolbachia* coverage was low, such in *favelas* where access was difficult, dengue infections were still reduced by 38% and chikungunya by 10% [12]. Another cluster randomized trial is underway in Belo Horizonte, Brazil.

In Colombia, pilot wMel *Wolbachia* releases began in the city of Bello in the Aburrá Valley in 2015 and were expanded in 2017 to city-wide deployments throughout nearby Medellín, Itagüí and Bello. An evaluation based on routine disease surveillance data reported reductions in notified dengue incidence of 95% to 97% in the three cities following *Wolbachia* introduction compared to the prior decade; a parallel case-control study in Medellín also showed significantly lower dengue incidence in *Wolbachia*-treated neighborhoods compared to untreated ones [13–16]. Deployment progressed to Cali, with phased releases since 2020. In May 2023, Cali's coverage reached 50% and the departmental and municipal governments announced the expansion of *Wolbachia* to Yumbo municipality, 13 km northeast of Cali [17].

*Wolbachia* is predicted to be a highly cost-effective intervention for controlling mosquito-borne illnesses, especially when released in high-density urban areas. In Indonesia, *Wolbachia* was projected to have a cost-effectiveness ratio in US dollars (US$) of US$1,500 per disability-adjusted life year (DALY) averted, offsetting much of the costs to the health system and to society with benefit-cost ratios ranging from 1.35 to 3.40 [18]. A study in Vietnam found the technology similarly cost effective using a 10-year time horizon and cost-saving at a 20-year time horizon [19]. A simulation across seven Brazilian cities also found *Wolbachia* cost-effective across all seven cities modeled and cost saving in five of them [20]. In Suva, Fiji, a smaller city, *Wolbachia* was acceptably cost-effective, but in Port Vila, Vanuatu, the very small target population and lower population density would not make the approach cost-effective there [21]. A simulation for Thailand suggested that *Wolbachia* combined with vaccination could be cost-effective [22].

To inform decision making within Colombia, we modeled the large-scale implementation of the *Wolbachia* replacement strategy for controlling dengue in 11 target Colombian cities. Here we present the resulting cost-effectiveness and benefit-cost analyses.

## Methods

### Framework

The WMP identified 11 target Colombian cities that might be suitable for *Wolbachia* based on population size, population density, and dengue incidence rates, and provided information about each city (S1 Table). The cities are distributed across

the western and central parts of Colombia, as shown in Fig 1. All together, these cities accounted for a third of Colombia's reported dengue cases from 2010 through 2019. Colombia's largest city and capital, Bogotá, is virtually free of dengue due to its high altitude, so it was not a target city.

The analyses were done by city, as costs, impacts, and funding decisions lie partly at the municipal level. The replacement strategy is viable only in areas with a sufficiently high population density to sustain the *Wolbachia*-infected mosquitoes (e.g., at least 1,000 inhabitants per km$^2$) [18]. The WMP defined the potential release area of each target city and calculated its population. Implementation was assumed to entail release of wMel infected mosquitoes (the most widely used *Wolbachia* strain) using the replacement strategy in the release area.

Our analysis began by estimating the baseline situation in the release area of each of these target cities in the absence of *Wolbachia*. Baseline variables included the average annual numbers of cases, health care costs, and loss of health from dengue cases. We then modeled the impact of a wMel *Wolbachia* replacement program in the release area of each target city based on the Yogyakarta cluster-randomized trial. Next, we examined the cost of implementing *Wolbachia* based on the WMP's recent Colombian projects. Finally, we calculated cost-effectiveness and benefit-cost ratios showing the ratio of estimated costs to predicted health care gains by city.

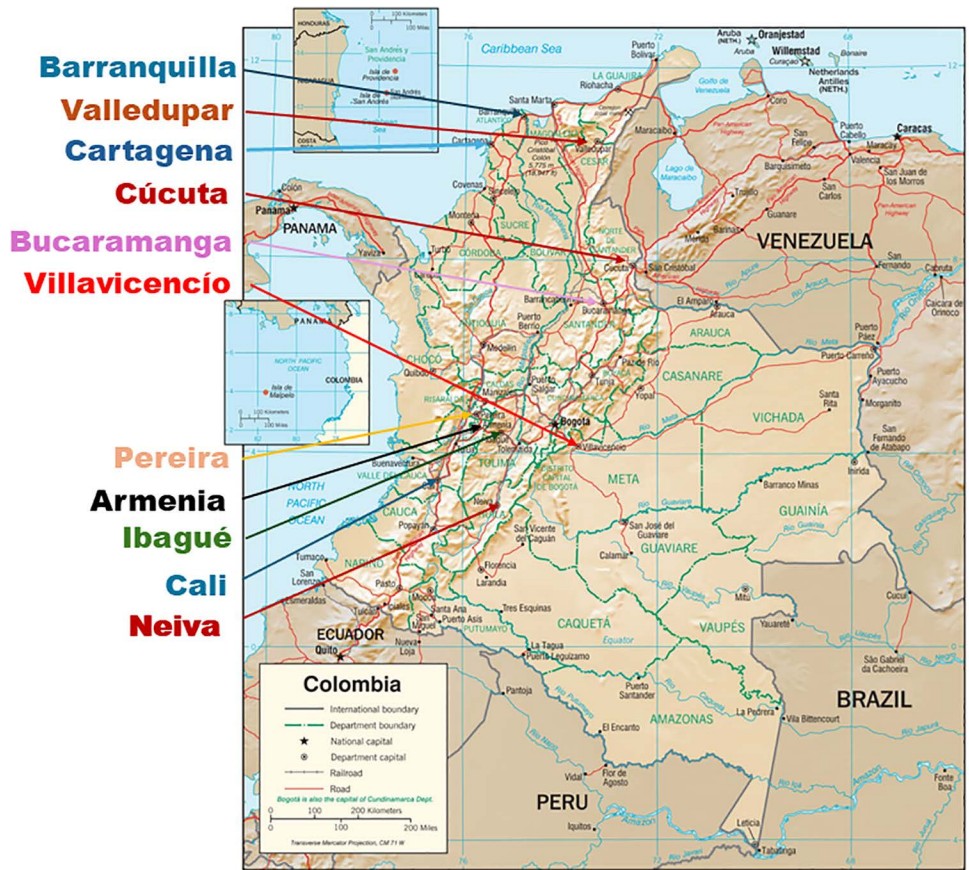

**Fig 1. Map of Colombia showing the 11 target cities.** The base map was reprinted from https://www.cia.gov/resources/map/colombia/ under a CC BY 4.0 license. The base map is in the public domain.

 

## Parameters

Table 1 provides the necessary national parameters for the economic analysis and their sources.

Items used to derive P1 and P2 were based on 2019 values, the last pre-pandemic year with complete data. The reliance on data from 2019, rather than subsequent years, was based on Colombia's experience with the COVID-19 pandemic. During the pandemic, staff were afraid to work, and patients minimized visits to health facilities for fear of contracting COVID-19. Therefore, later (pandemic) years would not have provided a representative foundation for future projections. Values from 2019 were adjusted for inflation and changes in exchange rates, giving virtually identical values in 2020 US$ as of 2019.

Parameter P3, cost data, were provided by the WMP based on its budget projections. These give representative projections for all target cities for future non-pandemic periods. In Cali, Phase I implementation had already begun as Colombia was affected by the pandemic. As the pandemic struck, workers could no longer move freely around the city. The resulting delays due to interruptions from the COVID-19 pandemic increased costs. Our projected costs for Cali across all phases of US$96,698 per square km factored in these COVID-19-related delays.

For parameters P4 through P8, findings from Yogyakarta and *Wolbachia's* characteristics support the projected reductions in conventional vector control expenditures as *Wolbachia* are introduced. After *Wolbachia* were released in Yogyakarta's intervention clusters, its District Health Office limited focal spraying of insecticide around the residence of notified

**Table 1. National Parameters[a].**

| Label | Parameter and data source | Value |
|---|---|---|
| P1 | Average health system cost per dengue case in 2019–20 for cases treated in the medical sector, US$ (see Results section) | $202.11 |
| P2 | Average health system cost per dengue case in 2019–20 for cases treated in the medical and non-medical sectors combined, US$ (see Results section) | $116.90 |
| P3 | Estimated cost of *Wolbachia* per km$^2$ in target cites in Colombia, US$ (from budget projections of WMP) | $87,625 |
| P4 | Estimated % savings in conventional vector control spending, year 1 (based on Yogyakarta experience) | 0% |
| P5 | Estimated % savings in conventional vector control spending, year 2 (based on Yogyakarta experience) | 20% |
| P6 | Estimated % savings in conventional vector control spending, year 3 (based on Yogyakarta experience) | 30% |
| P7 | Estimated % savings in conventional vector control spending, year 4 (based on Yogyakarta experience) | 40% |
| P8 | Estimated % savings in conventional vector control spending, years 5+ (based on Yogyakarta experience) | 50% |
| P9 | Efficacy of *Wolbachia* intervention (%), year 1 from date of deployment (see section "Effectiveness of a *Wolbachia* program") | 37.5% |
| P10 | Efficacy of *Wolbachia* intervention (%), years 2+ from date of deployment (see section "Effectiveness of a *Wolbachia* program") | 75.0% |
| P11 | Efficacy of *Wolbachia* intervention (%), 10-year average from date of deployment (weighted average of P9 for 1 year and P10 for 9 years) | 71.3% |
| P12 | Annual discount rate for costs and health effects [23] | 3% |
| P13 | DALY/dengue case (see "Parameters") | 0.0476 |
| P14 | Share of *Wolbachia* deployment cost that is incurred in year 1 (see "Cost of *Wolbachia* deployment") | 100% |
| P15 | Share of *Wolbachia* deployment cost needed for long term monitoring, year 2+ (see "Cost of *Wolbachia* deployment") | 1% |
| P16 | Cumulative present value factor over 10 years using P12 (see "Parameters") | 8.53 |
| P17 | Colombia GDP/capita (2020), World Bank, market prices, US$ [25] | $5,312 |
| P18 | Share of dengue cases correctly reported in the surveillance system (from our expert panel) | 29% |
| P19 | Share of *Wolbachia* program costs for preparation before deployment (see "Cost of *Wolbachia* deployment") | 20.54% |

[a]DALY denotes disability-adjusted life year; GDP denotes gross domestic product; km denotes kilometers; WMP denotes World Mosquito Program. Monetary amounts are in 2020 United States dollars (US$) at market exchange rates. Throughout this analysis, "national" refers to the aggregate of all target cities.

dengue cases "subject to resource availability and local transmission risk"[9]. The district's fogging rate proved 83% lower (95% confidence interval 70%-90%) in *Wolbachia*-treated areas compared to untreated areas. *Wolbachia*-infected mosquitoes pose minimal risk of causing dengue illness, while fogging in such areas would have killed these helpful *Wolbachia*-infected mosquitoes and might have promoted the introduction of harmful, wild mosquitoes.

For parameter P12, we relied on a leading textbook on economic evaluation in health [23]. For P13, the disease burden per case of dengue is the sum of its morbidity and mortality components. The morbidity component was 0.032 [24]. The mortality component was calculated first by dividing the average number of deaths due to dengue between the years 2012–2018 by the average incidence for these same years. The resulting weighted average case-fatality rate was $6.05 \times 10^{-4}$. Based on an estimated 50 years of remaining life (as young adults are the median age of dengue fatalities) and P12, the discounted remaining life was 25.73 years, calculated as: $[1 - (1+P12)^{-50}]/P12$. The mortality component was 0.0156 DALYs (i.e., $6.05 \times 10^{-4}$ x 25.73). The overall burden per case was 0.0476 DALYs (i.e., $0.032 + 0.0156$).

Parameter P16, the cumulative present value factor (8.53) was calculated with the Excel present value function (PV) using P12 and a time horizon of 10 years, i.e., PV(P12,10,-1).

Parameter P17 is from the World Bank [25]

## Disease burden of dengue

In Colombia, as in most countries in which dengue is endemic, facilities and providers in the formal health care sector are supposed to report each suspected dengue case to the *Sistema Nacional de Vigilancia en Salud Pública* (SIVIGILA) [National Public Health Surveillance System] [26]. At the start of the planning process, the WMP downloaded the number of reported dengue cases from the populated portion of each target city from 2010 through 2019 and calculated the annual average. This averaging was done to provide a stable, representative estimate of the annual number of future cases expected with no intervention. As dengue is a communicable infectious disease, its incidence varies several fold from one year to the next, so data for the latest year would not have provided a representative basis of future planning.

The disease burden of dengue (loss of good health) in a specified geographical area in a year is best conceptualized as the product of its number of dengue cases times the disease burden per case. Like most national systems, SIVIGILA is a passive surveillance system. A dengue episode is counted only if the patient visits a formal healthcare provider that submits data to SIVIGILA, the provider classifies the episode as dengue based on available clinical, laboratory, and epidemiologic data, AND the provider enters the case into SIVIGILA. Global research has found that a substantial share of dengue cases are treated outside the formal health sector and thus not captured in existing databases [27]. To apply this concept to Colombia, we assessed the breakdown of dengue cases by severity and reporting status. We relied on the expertise of three epidemiologists: Luz Inés Villarreal Salazar (independent consultant in Colombia), co-author Carlos Willian Rincon Perez (University of the Andes), and Maria Patricia Arbelaez Montoya (World Mosquito Program, Colombia). We adjusted for underreporting of the number of dengue cases using an adjustment factor derived from *SIVIGILA* and *Registro Individual de Prestación de Servicios de Salud (RIPS)* [Individual Registry of Provision of Health Services]. The expert panel recognized that some common dengue symptoms, such as fever and discomfort, can be caused by other infections as well. Therefore, some dengue illnesses might be mistakenly attributed to other causes. In parameter P18, the panel estimated the remaining share, after deducting the misclassifications, which was correctly reported as dengue.

## Effectiveness of a *Wolbachia* program

To adjust for the fact that routine programs often have fewer resources and less intensive supervision than research trials, we rounded down the per-protocol efficacy from the Indonesian cluster-randomized trial [8]. When a target city is chosen in our modeling study, we assume that planning occurs in year 0 (the selection year) and release of adult mosquitoes begins in the first year (termed year 1). We projected that the *Wolbachia* program in Colombia will result in a 75%

reduction in dengue cases once *Wolbachia* is stably established in the mosquito population--the second year of implementation onwards from the projected time for deployment.

Projecting a linear increase from zero to complete establishment of *Wolbachia* over the first year of deployment, we estimated a 37.5% (half-way) reduction in dengue cases overall in the first year of *Wolbachia* deployment. The Data Availability statement provides further details on the external Colombian databases.

## Cost of a dengue episode

A city's aggregate cost of dengue is the product of the average cost per case times its number of cases. We used two approaches to estimate the cost of a dengue case in Colombia. Under our main approach, the average direct cost of a dengue case treated in the formal health system in 2019 was estimated using the tariffs to pay treatment costs from transit accidents, *Seguro Obligatorio para Accidentes de Tránsito (SOAT)* [Compulsory Insurance for Traffic Accidents]. Because car insurance is mandatory in Colombia and SOAT is operated by Colombia's national government and reports publicly, the SOAT tariffs also serve as reference prices in payment negotiations between insurers and providers [28]. While actual payment rates from other insurers are not publicly available, experts believe that actual payment rates likely average the SOAT rates. Anecdotal reports suggest that in rural areas, where providers are few, providers command payments above the SOAT rates, whereas in urban areas, where providers are numerous, payers can negotiate discounts below the SOAT rates.

We converted the SOAT amounts in Colombian pesos to US dollars at the average exchange rate for the years 2015–2020 [29]. For most curative services in the health care system, RIPS provides a national claims system that captures the health care provided to the insured population by diagnostic codes, care provided, and care setting. The data include the number of consultations and procedures used, emergency room visits, and hospitalizations. RIPS categorized dengue cases as classic dengue and severe dengue. For verification we used the *Suficiencia* [Sufficiency] database, which provides service payments for calculating the *Unidad de Pago por Capitación (UPC)* [Capitation Payment Unit] and premium information [30].

Using the SOAT tariff, we derived the cost per case for each severity level of dengue. We then calculated a weighted average based on the estimated share of dengue cases by severity. Using severity level, rather than treatment setting, allowed us to incorporate surveillance data, which has severity but not setting. To report the cost of all types of dengue cases in Colombia from the health system perspective, we adjusted for cases treated outside the health care system. To estimate the economic cost, we incorporated both the cost of cases treated outside the health care system and direct and indirect household expenditures during a dengue episode. We then analyzed the RIPS utilization data to derive the average cost of a non-fatal dengue case for the years 2015–2020 and reported the average 5-year cost per case based on the severity of dengue, i.e., severe and non-severe dengue. The RIPS data included the number of dengue health care services based on the care setting: consultations, procedures, emergencies, and hospitalizations.

All economic data sets used in this study were public anonymous data files for computing counts and means. As further protection of anonymity, it was not possible to link a service with a specific provider, a patient's identification number, nor other services received by the same patient. To avoid errors, all extractions were done by the co-author who uses the databases regularly (CWRP).

## Validation of cost per dengue case

To validate our SOAT-based estimate of the healthcare cost per dengue case, we used aggregate data (see S1 Text, S2 Table and S3 Table) [31–36]. This aggregate approach, termed macro-costing, uses the aggregate cost of a hospital or system of hospitals to derive key unit costs. These are the average cost of a hospitalization and of an ambulatory visit. Macro-costing performs this calculation by converting the entire output (annual services) of the hospital

or system in terms of bed-day equivalents. The conversion is based on a systematic review that found an average an ambulatory visit used 0.32 times the resources of an average inpatient day [35]. The validity of the approach for hospitalizations rests on the assumption that the resource use of an average inpatient day for dengue is comparable to the average resource use for all inpatient days. Similarly, the validity of the approach for ambulatory care rests on the assumption that the resource use of an average ambulatory visit for dengue is comparable to the average resource use for all ambulatory days.

Two offsetting factors support the validity of macro-costing estimates for dengue. As dengue management requires no surgery and few specialized drugs or laboratory tests, it should be less expensive than corresponding services overall. On the other hand, dengue is usually an acute condition requiring shorter hospitalizations and fewer visits than many other types of illness. As the start of treatment for most illness episodes tends to involve the most examinations, shorter episodes would tend to have higher resource use per visit or per day than longer episodes. In view of the many uncertainties, however, macro-costing remained a secondary estimate for assessing the potential cost offsets from dengue prevention.

### Disease burden of dengue per case

Based on the calculation provided for discounted remaining life, the years of life lost and years lost to disability per case are 0.0156 and 0.0320, [24] corresponding to shares of 33% and 67%, respectively. Their sum (0.0476 DALYs) comprised the total disease burden per dengue case.

### Cost of *Wolbachia* deployment

To estimate the cost of the *Wolbachia* program in each of the 11 target cities in Columbia, we started by analyzing the program budget for Cali. The budget covered two programmatic phases, with each phase divided into three stages: prepare, release, and short-term monitoring (STM). The budget covers the administrative and management cost, communication, community engagement, data management, diagnostic, monitoring, mosquito rearing, the release of the *Wolbachia* mosquitoes, surveillance, site start-up, project oversight, and indirect (facilities and administrative) costs. As the only one of the target cities with funding mechanisms to implement the first phase of its plan, Cali's budget provides a realistic blueprint. Following the successful approach from Bello, Medellín and Itagüí, Cali's program raises *Wolbachia*-infected mosquitoes in an insectary and then releases the adult mosquitoes from a vehicle.

In Phase 1, Cali's preparation stage lasted 12 months, release took 6 months, and the STM was planned for 12 months for a total of 30 months. Initially, the WMP projected that implementation of the *Wolbachia* program would take about 30 months per city. After further review, however, WMP officials and the authors agreed that experience to date would allow an accelerated timeline in future cities, spending virtually 100% of the budget in the first year, cutting the overall projected time requirement to 15 months per city and reducing staff costs. This accelerated timeline generated the share of costs in the preparatory phase for planning and public communication, prior to deployment. In reviewing the costs from Cali, we also noted that some items were one-time expenses due to the need to pause deployment due to the COVID-19 pandemic. Such items would not be expected in future cities.

We estimated the indirect (facilities and administrative) cost of the *Wolbachia* program at 15% of direct costs. This is the maximum global rate allowed to grantees by the Bill & Melinda Gates Foundation, [31] a major sponsor of *Wolbachia* development. Incorporating the lower indirect cost and adjusting staff months based on the shortened time frame, we derived an adjusted cost per km$^2$ (parameter P5). WMP estimated the projected release area (km$^2$) in each target city, including all built-up areas and excluding public spaces, parks, and empty spaces. This area was multiplied by the adjusted cost per km$^2$ (parameter P3) to estimate the cost of implementation in the rest of Cali and the 10 other target cities. We projected an estimated 1% of the initial spending needed annually for long-term monitoring from the second year onward.

## Medical cost offsets

We calculated the medical cost offsets from dengue cases averted each year as the cost per symptomatic case times the baseline average number of such cases times the fraction averted in each city year. Although some health economists disagree with discounting future health effects, [37] a leading textbook and Colombia-specific guidelines recommend that future costs and health benefits should be discounted [23,38]. As our base case, we calculated the present value of the *Wolbachia* program and all cost offsets in each city over a ten-year time horizon with a discount factor from P12. The vector control offset was calculated through percentagewise cost savings in parameters P4 through P8. The medical cost offset comprises the estimated reduction of cases over the ten-year time horizon. The present value of these offsets was calculated as the annual full-deployment result times the cumulative present value factor for ten years (parameter P16) less an adjustment for the smaller effectiveness in year 1.

## Economic appraisal

To value the indirect benefits (gains in quality and length of life), we needed to assign an economic value to a year of good health—averting a DALY or gaining a Quality-adjusted Life Year (QALY). This valuation is equivalent to setting a threshold value for determining the cost-effectiveness of a health intervention. In 2001, the World Health Organization's Macroeconomic Commission on Health recommended thresholds of 1 and 3 times a country's per capita Gross Domestic Product (GDP) for an intervention to be "very cost-effective" or "cost effective," respectively [39]. Subsequently, WHO officials recommended finding evidence-based thresholds and incorporating fairness and affordability into the decision process [40]. Economic theory suggests that evidence consider the public's willingness to pay (WTP) to avert one DALY or gain one QALY [23].

To apply this concept, we searched PubMed for studies on WTP in Colombia. The one study we found, modeling chemotherapy for lung cancer, did not present an empirical estimate, but simply selected a value of US$17,656, three times Colombia's then GDP per capita [41]. Broadening the search to a global review of WTP studies, found a median value for upper-middle income countries (the relevant category for Colombia) of US$5,936, with an interquartile range of US$7,233 [42]. However, none of the included studies was conducted in Colombia and upper-middle income countries span a wide range of per capita GDP. However, in 2023, an empirical approach was published for WTP thresholds and applied to 174 countries, including Colombia [43]. Based on national data rather than survey responses, it calculated national WTPs based on the country's changes in life expectancy and health expenditures. This approach found that Colombia's WTP per QALY gained (equivalent to a DALY averted) was 0.75 times its per capita GDP in 2019. An independent commentary noted the many advantages of this approach [44]. Like that in most upper-middle income countries, Colombia's WTP as a proportion of its per capita GDP fell in the range of 0.5 to 1.0. Applying Colombia's ratio, we valued each DALY in our target year (2020) in Colombia as 0.75 times that year's per capita GDP. That is, Colombia's WTP threshold is US$3,984 (0.75 times Colombia's per capita GDP [21] of US$5,312). Thus, each DALY averted through reduced dengue had an economic value of US$3,984.

We calculated each city's benefit-cost ratio as its total economic benefits (including the economic value of good health) divided by the cost of the deployment. If this ratio exceeded 1.0, *Wolbachia* was considered a favorable economic investment. The incremental cost-effectiveness ratio (ICER) is the net present value cost of the *Wolbachia* program divided by its present value health gain in DALYs. A positive ICER below Colombia's threshold indicates that the intervention is cost-effective. A negative ICER indicates that the replacement strategy is cost saving in that city, i.e., exceptionally cost-effective.

## Sensitivity analyses

Keeping our central discount rate at 3% per year (parameter P12), we performed sensitivity analyses for Cali at alternative discount rates of 0% and 6% per year. Different discount rates were expected to have little impact on the present value

of program costs, which occur primarily at the start, but would affect the present value of offsets to health care and vector control costs and the present value of the economic value of improved health.

### Ethics Statement

This modeling study did not involve any human studies data as it was based entirely on aggregate or publicly available anonymous data. These data could not allow any individual to be identified nor linked with any individual. The research team did not prospectively nor retrospectively recruit human participants, nor did the team obtain tissues, data, or samples for the purposes of this study. The research team did not review existing medical records nor archived samples. Therefore, this study was outside the purview of the Committee for Protection of Human Studies in Research so ethical approval was not applicable.

## Results

### Current cost per case of dengue

The epidemiological panel and SIVIGILA data distributed dengue cases in Colombia by severity and reporting into five categories (see S4 Table): (1) 1.87% are severe cases and correctly diagnosed and reported to SIVIGILA, (2) 27.13% are non-severe dengue (including those with and without warning signs) and correctly reported to SIVIGILA, (3) 11% are non-severe dengue, diagnosed by medical providers but not reported to SIVIGILA due to time and administrative barriers, (4) 20% are non-severe dengue cases that are misdiagnosed (e.g., diagnosed as a non-specific viral fever), and (5) 40% are mild and do not interact with the formal healthcare system (i.e., home treatments).

Our panel estimated that only 29% of dengue cases are reported, almost all of which are non-severe dengue. Based on SOAT tariff, we estimated the healthcare cost of care for cases within the medical system as US$406.37 for a severe case (constituting 6.45% of medical cases) and US$188.02 for a non-severe medical dengue case (constituting 93.55% of medical cases). The weighted average healthcare cost per medical case was US$202.11 and US$1.50 for a non-medical dengue case. The overall health sector costs per case (including non-medical cases) averaged US$116.90 based on the SOAT tariff and $US$121.02 based on macro-costing.

Table 2. Aggregate demographic and epidemiologic data for target cities[a].

| Rank | Municipality | Adjusted population in release area | Average notified release area dengue cases | Projected release area dengue cases without treatment |
|---|---|---|---|---|
| 1 | Cali | 2,217,961 | 8,018 | 27,649 |
| 2 | Ibagué | 503,745 | 2,999 | 10,342 |
| 3 | Villavicencio | 506,145 | 2,947 | 10,161 |
| 4 | Cúcuta | 759,395 | 2,824 | 9,739 |
| 5 | Bucaramanga | 604,186 | 2,767 | 9,540 |
| 6 | Neiva | 343,194 | 2,040 | 7,035 |
| 7 | Barranquilla | 1,296,471 | 1,744 | 6,015 |
| 8 | Valledupar | 477,763 | 1,142 | 3,937 |
| 9 | Armenia | 300,785 | 1,189 | 4,100 |
| 10 | Pereira | 404,270 | 946 | 3,262 |
| 11 | Cartagena | 926,747 | 713 | 2,460 |
| ALL | National | 8,340,662 | 27,329 | 94,239 |

[a]Note: population in the release areas derived by the World Mosquito Program based on analyses of population density; monetary amounts are in 2020 US dollars.

Cities are ranked in decreasing number of average annual dengue cases from 2010 through 2019 (see S1 Table).

**Table 3. Aggregate costs and DALYs for target cities following the start of *Wolbachia* releases[a].**

| Rank | Municipality | Cost of dengue cases[b] | Cost of regular vector control (release area) | Initial year *Wolbachia* deployment costs | PV *Wolbachia* program costs[c] | PV vector control offsets[c] | PV medical cost offsets[c] | PV net costs[c] | PV DALYs[a] |
|------|--------------|-------------------------|-----------------------------------------------|--------------------------------------------|----------------------------------|------------------------------|----------------------------|------------------|-------------|
| 1 | Cali | $3,232,127 | $169,567 | $8,973,571 | $9,672,263 | $563,261 | $20,086,318 | –$10,977,315 | 8,174 |
| 2 | Ibagué | $1,208,900 | $71,912 | $2,269,484 | $2,446,189 | $238,873 | $7,512,810 | –$5,305,494 | 3,057 |
| 3 | Villavicencio | $1,187,816 | $93,172 | $2,506,072 | $2,701,197 | $309,493 | $7,381,782 | –$4,990,078 | 3,004 |
| 4 | Cúcuta | $1,138,471 | $359,897 | $4,363,719 | $4,703,483 | $1,195,491 | $7,075,123 | –$3,567,131 | 2,879 |
| 5 | Bucaramanga | $1,115,217 | $241,890 | $1,989,085 | $2,143,957 | $803,501 | $6,930,606 | –$5,590,150 | 2,821 |
| 6 | Neiva | $822,322 | $47,996 | $1,857,647 | $2,002,286 | $159,430 | $5,110,385 | –$3,267,529 | 2,080 |
| 7 | Barranquilla | $703,192 | $68,121 | $5,783,242 | $6,233,532 | $226,281 | $4,370,044 | $1,637,207 | 1,778 |
| 8 | Valledupar | $460,213 | $88,882 | $2,234,434 | $2,408,410 | $295,246 | $2,860,029 | –$746,866 | 1,164 |
| 9 | Armenia | $479,300 | $37,526 | $1,253,036 | $1,350,598 | $124,653 | $2,978,651 | –$1,752,705 | 1,212 |
| 10 | Pereira | $381,265 | $38,458 | $1,524,673 | $1,643,386 | $127,747 | $2,369,401 | –$853,762 | 964 |
| 11 | Cartagena | $287,532 | $332,320 | $3,864,257 | $4,165,132 | $1,103,887 | $1,786,889 | $1,274,356 | 727 |
| ALL | National | $11,016,355 | $1,549,740 | $36,619,221 | $39,470,433 | $5,147,863 | $68,462,037 | –$34,139,468 | 27,862 |

[a]Note: DALYs = disability adjusted life years; PV = cumulative present value over 10 years discounted using P12; population in the release areas derived by the World Mosquito Program based on analyses of population density; monetary amounts are in 2020 US dollars. Cities are ranked in decreasing number of average annual dengue cases from 2010 through 2019 (see S1 Table)

[b]10-year present values. [c]Cost of dengue cases (reported, unreported but treated in medical setting, and cases treated in non-medical settings).

## Economic results in target cities with a 10-year horizon

Tables 2 and 3 display the core analytic results of *Wolbachia* releases for each target city and the national total (sum of all target cities). Table 3 shows that initial year cost of *Wolbachia* substantially exceeds the annual cost of conventional vector control. At the national level, this *Wolbachia* cost ($36,619,221) is 24 times the conventional cost ($1,549,740).

Table 4 presents the costs and benefits as rates per person covered and gives the ICER and other economic results. All ICERs are below the threshold of US$3,984, with the highest (US$1,752 for Cartagena) still only 0.44 times this threshold. For 9 of the 11 cities and national (all-cities) value, the ICERs are negative, indicating that the *Wolbachia* would be cost saving individually in those cities and nationally. All benefit-cost ratios are favorable or highly favorable (exceed 1.00), ranging from 1.39 to 8.85.

## Effect of alternative time horizons

To illustrate our results in greater detail, we have focused on Cali, the city with the greatest burden in reported dengue cases. After the Aburrá Valley, where *Wolbachia* had been deployed previously, [13–16] Cali is the one target city in which *Wolbachia* is already partly deployed. Fig 2 displays the cumulative projected economic benefits of the *Wolbachia* program in Cali by component and time horizon, where time is the number of completed years since *Wolbachia* deployment. *Wolbachia* is projected to replace some conventional vector control, lower the need for medical care for treating dengue illness, and create economic value of additional healthy years. The overall economic benefits, the sum of these components, grows with increasing time horizons to US$42.97 per person covered with a 20-year horizon. Over this horizon, indirect benefits (the economic value of reduced illness, US$26.27) are the largest component, followed by medical care offsets (US$16.20), with vector control offsets as the smallest benefit (US$0.50).

In Fig 3 the upper (dashed red) line shows the cost per person of implementing the *Wolbachia* program. This starts in year 0 with 20.54% of initial program costs (US$0.83) for planning and engagement of residents and local leaders. In year 1, the year in which city-wide releases would occur, the remainder of initial costs occur, bringing initial program costs to US$4.05. Thereafter, annual monitoring occurs, costing 1% of the initial costs annually throughout the remainder of the

**Table 4. Ratios for target cities following the start of *Wolbachia* releases[a].**

| Rank | Municipality | PV *Wolbachia* deployment costs per person covered | PV conventional vector control offsets per person covered | PV medical care offsets per person covered | PV indirect benefits per person covered | PV overall gross benefits per person covered | PV DALYs averted per 100,000 population | Benefit-cost ratio | ICER |
|---|---|---|---|---|---|---|---|---|---|
| 1 | Cali | $4.36 | $0.25 | $9.06 | $14.68 | $23.99 | 369 | 5.50 | -$1,343 |
| 2 | Ibagué | $4.86 | $0.47 | $14.91 | $24.18 | $39.57 | 607 | 8.15 | -$1,735 |
| 3 | Villavicencio | $5.34 | $0.61 | $14.58 | $23.65 | $38.84 | 594 | 7.28 | -$1,661 |
| 4 | Cúcuta | $6.19 | $1.57 | $9.32 | $15.11 | $26.00 | 379 | 4.20 | -$1,239 |
| 5 | Bucaramanga | $3.55 | $1.33 | $11.47 | $18.60 | $31.40 | 467 | 8.85 | -$1,982 |
| 6 | Neiva | $5.83 | $0.46 | $14.89 | $24.14 | $39.50 | 606 | 6.77 | -$1,571 |
| 7 | Barranquilla | $4.81 | $0.17 | $3.37 | $5.47 | $9.01 | 137 | 1.87 | $921 |
| 8 | Valledupar | $5.04 | $0.62 | $5.99 | $9.71 | $16.31 | 244 | 3.24 | -$642 |
| 9 | Armenia | $4.49 | $0.41 | $9.90 | $16.06 | $26.37 | 403 | 5.87 | -$1,446 |
| 10 | Pereira | $4.07 | $0.32 | $5.86 | $9.50 | $15.68 | 239 | 3.86 | -$885 |
| 11 | Cartagena | $4.49 | $1.19 | $1.93 | $3.13 | $6.25 | 78 | 1.39 | $1,752 |
| ALL | National | $4.73 | $0.62 | $8.21 | $13.31 | $22.13 | 334 | 4.68 | -$1,225 |

[a]Note: DALYs = disability adjusted life years; ICER = incremental cost-effectiveness ratio; PV = present value over 10 years discounted from P12. Monetary amounts are in 2020 US dollars. Cities are ranked in decreasing number of average annual dengue cases from 2010 through 2019 (see S1 Table). National represents the sum of all target cities.

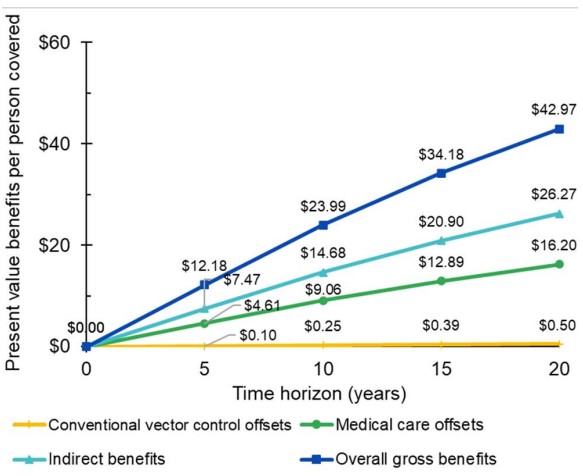

**Fig 2. Economic benefits of *Wolbachia* by component and time horizon.**

time horizon. Thus, cumulative present value *Wolbachia* implementation costs per person rise to US$4.20 through 5 years and US$4.63 through 20 years. Because deployment costs occur early (mostly in year 1) and monitoring costs are relatively small, longer time horizons have little impact on the present value of *Wolbachia* program costs.

The lower (blue) line is the net healthcare cost at each time horizon. In year 0, when there are no offsets, it is identical to costs of planning and engagement (US$0.83). In year 1, with all deployment costs incurred but little conventional vector

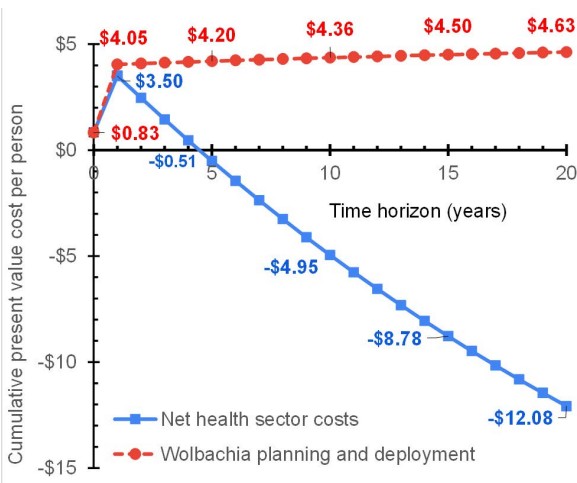

**Fig 3. Costs by component and time horizon.** Net health sector costs are costs of *Wolbachia* planning and deployment less conventional vector control offsets and medical care offsets.

control and medical care offsets, net present value healthcare costs per person covered reached the maximum (US$3.50). In subsequent years, healthcare offsets exceed the additional vector control costs. Beyond 4.3 years, *Wolbachia* becomes cost saving in health care costs. With longer time horizons, the cost offsets continue to grow. Net costs per person become substantial negative numbers (negative US$4.95 and US$12.08) at the 10- and 20-year horizons, respectively.

Fig 4 shows the summary outcome measures on health (DALYs averted) and economic impact (benefit-cost ratio) for Cali. Both measures increase with longer time horizons. With a 10-year horizon, the *Wolbachia* program averts 369 DALYs per 100,000 population with a benefit-cost ratio of 5.50. This highly favorable 10-year ratio indicates that every dollar invested generates US$5.50 in economic benefits for the city's residents through better health and averted healthcare costs.

With a 20-year horizon, these results become almost twice as favorable, averting 659 DALYs and a benefit-cost ratio of 9.29 to 1. Since the economic benefits from better health and offsets to health care expenses occur approximately uniformly over time, the break-even time horizon at which the overall economic benefits exactly offset the costs is only 1.72 years (21 months) in Cali.

**Effect of alternative discount rates.** Our sensitivity analysis showed that with no discounting (i.e., 0%), the 10-year benefit-cost ratio increased to 6.24 while it fell to 4.89 with a higher (6%) discount rate, the benefit-cost ratio fell to 4.89 (data not shown in tables). Both results are highly favorable. Because higher discount rates lower both the numerator (economic value of health improvements) and denominator (net cost after offsets from savings in health costs), its impact on the benefit-cost ratio was relatively limited.

**National projections.** Extending our results nationally, Fig 5 presents the benefit-cost ratios for all target cities based on the 10-year horizon. Panel A displays the cities in decreasing order. Projections for all target cities are favorable, as all the ratios exceed 1.00. Cali is close to the national average. Cartagena is the most marginal in economic terms (ratio 1.39), while Bucaramanga, with a ratio of 8.85, is almost twice as favorable as the national average.

Panel B shows a scatter plot of these ratios in relation to population density and average annual dengue incidence. Higher values of both independent variables tend to be associated with higher (more favorable) benefit-cost ratios. Cities in the upper right corner are the most favorable, while those in the lower left corner (nearest the axis intersection) are less favorable. The two municipalities nearest the axis intersection, Barranquilla and Cartagena, are the only ones with positive

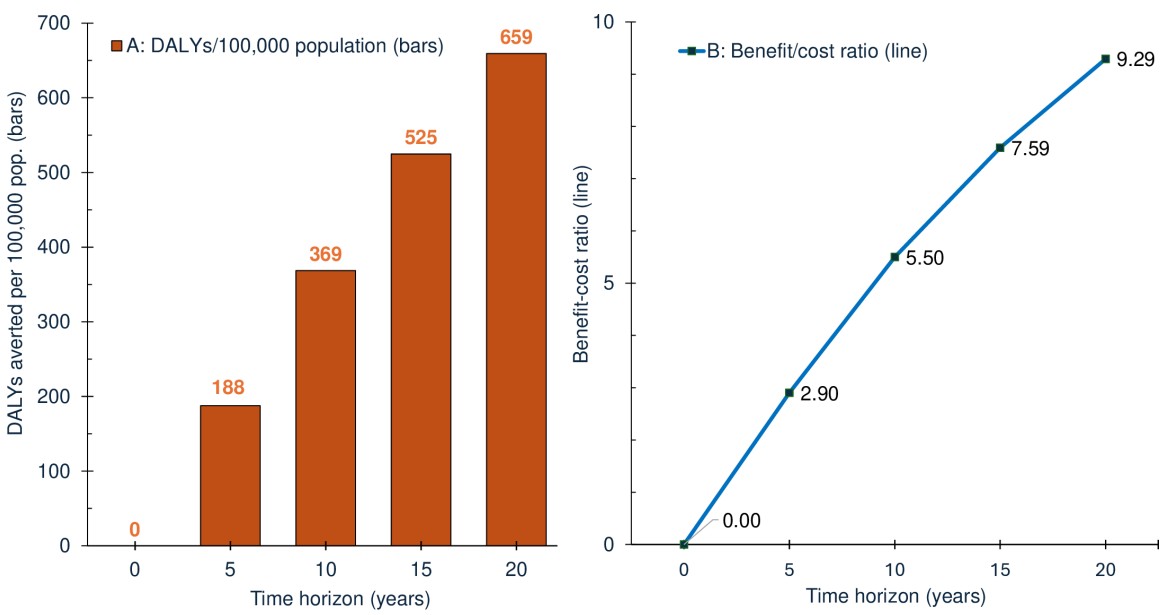

**Fig 4. Program impacts in Cali by time horizon: Disability adjusted life years (DALYs) averted (panel A) and benefit-cost ratios (panel B).**

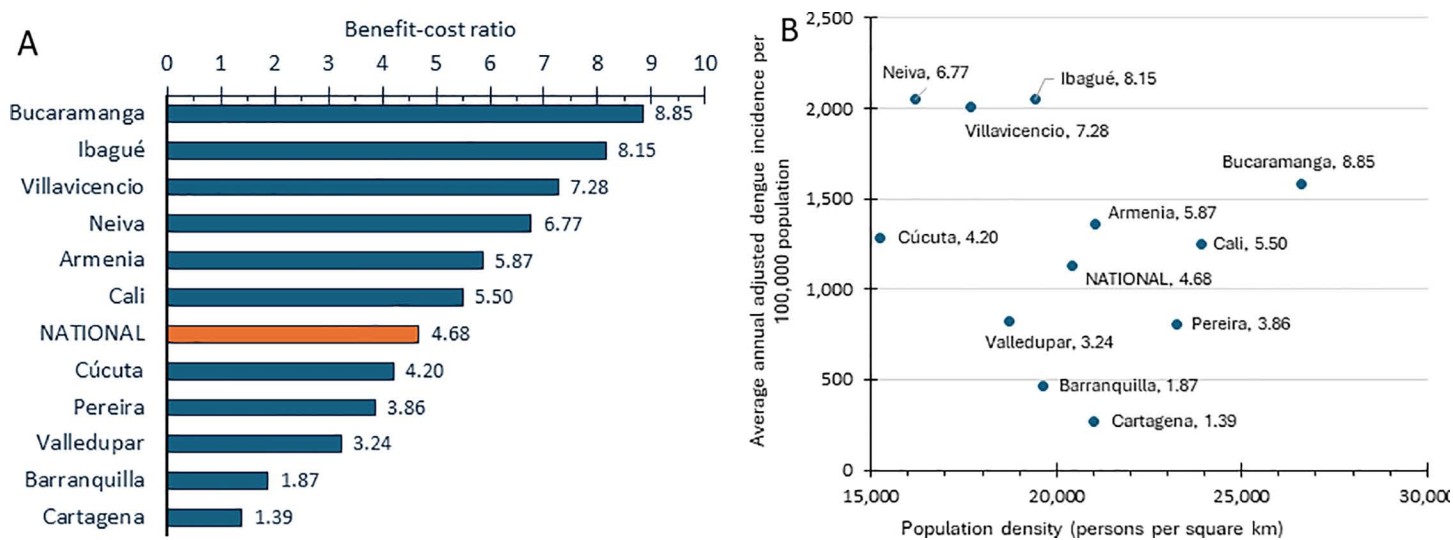

**Fig 5. Estimated benefit-cost ratios by city with a 10-year horizon.**

ICERs, indicating that *Wolbachia* would not be cost saving in those cities. Nevertheless, even in those cities the *Wolbachia* program was cost effective and cost beneficial. Dengue incidence, which varies 8-fold from the least to the most affected city, proved to be the more important determinant of the benefit cost-ratio. Higher population density, which varies by a factor of only 1.7, contributes only marginally to higher benefit-cost ratios.

## Discussion

Colombia is hyperendemic with dengue [2]. Accounting for cases treated outside the medical system, misdiagnosed, or otherwise not reported, we found that dengue incidence across all 11 target cities is 3.4 times the reported number. Our estimates reinforce previous research that Colombia's dengue burden per 100,000 population exceeds the global average [18,27].

If implemented with efficacy mirroring the results from the cluster-randomized trial, [8] *Wolbachia* will substantially mitigate dengue incidence in the target cities in Colombia. These impacts generate highly favorable benefit-cost ratios by averting healthcare costs and generating indirect benefits. In over half of the cities, including Cali, the 10-year economic benefits exceed US$5.00 for every dollar invested. While we project reductions in conventional vector control, these savings would offset only a small part of the *Wolbachia* program. The major economic benefits are averting medical care and better health.

In Cali, the most expensive activity was mosquito release, followed by mosquito rearing and community engagement. Managers may be able to lower future costs through economies of scale or identifying newer approaches. *Wolbachia's* costs mostly occur at the program's start, while the health and economic benefits accrue over time. Therefore, the cost effectiveness and economic benefits of *Wolbachia* improve with longer time horizons. For example, for each dollar invested, the benefit in Cali ranged from US$5.50 at 10 years to US$9.29 at 20 years. Thanks to Colombia's national health insurance system, the medical care component of these benefits would accrue largely to Colombia's public sector.

We validated the cost per case of dengue through a supplemental calculation using macro costing. The consistency between our main (SOAT) and supplemental approaches lent confidence in our results. The difference in cost per case between our main and supplemental approaches (US$116.90 and US$121.02, respectively) was only 3.5%. Because the SOAT approach provided greater detail, it was our preferred choice. We explored performing additional analyses by tier within Colombia's health system, but inconsistencies precluded doing this reliably with the available data (see S2 Text).

Global experience and models raise a caution that *Wolbachia* may not work in isolated circumstances. As one example, in two nearby sites in Vietnam, *Wolbachia* coverage dropped in one (Tri Nguyan village) but not in the other (Vinh Luong). Researchers speculated that elevated temperature in water storage tanks where mosquitoes bred or an interaction with the built environment may have inhibited *Wolbachia* replication in the ineffective village [45]. As another example, in small-scale releases in Malaysia, *Wolbachia* were not permanently established because the selected strain (wAlbB) may have been less fit than the wild mosquitoes [45]. Modeling studies raise the possibility that dengue viruses could become resistant to *Wolbachia*. Because of the multiple mechanisms by which *Wolbachia* inhibit dengue transmission, however, any such resistance, if any, would likely evolve only slowly [46]. Resistance could be identified by monitoring and possible corrective actions, such as new *Wolbachia* strains.

Any public health program risks interference from a major disruption, including war, political change, major budget cuts, or a pandemic. However, as *Wolbachia* bacteria are generally self-perpetuating once established, the program should be resilient. While such disruptions could delay expansion into new locations, they would have little effect on areas covered. As a *Wolbachia* program depends more on mosquito biology than on human behavior, we do not expect its efficacy to differ substantially between Indonesia and Colombia. To the extent that socio-economic levels may have some impact, these levels differ little between urban areas in the two countries. While Colombia's national GDP per capita in purchasing power parity in 2023 was about a third higher than Indonesia's (PPP$20,676 versus PPP$15.416), Colombia was more urbanized than Indonesia (82.35% versus 58.57% of the population was urban) [24]. Therefore, if, for example, the per capita GDP in each country's urban areas were twice that of its rural areas, we calculated that the per capita GDP in urban areas would only have been 17% higher in Colombia than in Indonesia. This calculation increases our confidence in adopting our conservative Indonesian efficacy for Colombia.

Our very favorable national benefit-cost ratio of 4.68 indicates that our findings are robust. Our calculations show that the replacement strategy would remain economically viable nationally even if 10-year efficacy declined relatively

by as much as 78.6%, calculated as (4.68–1)/4.68. With that large a decline, costs of US$1.00 would generate benefits of US$4.68 x (100.0% - 78.6%) or US$1.00, meaning that the program would just break even economically. Our projected 75% efficacy is likely conservative. As noted, wMel *Wolbachia* proved 95%-97% effective in a Colombian outcome study [15] and 82.7% effective in Yogyakarta after adjustment for border crossing [9]. A higher efficacy would raise the benefit-cost ratio.

While our benefit-cost ratios compare *Wolbachia* against no dengue control, policymakers may also wish to consider comparing *Wolbachia* against alternative dengue control strategies. An alternative vector control strategy based on community-based mobilization (*Camino Verde*) proved to be effective but labor intensive and expensive as originally implemented. Its cost-effectiveness ratios relative to GDP per capita were relatively unfavorable--3.0 in Mexico and 16.9 in Nicaragua [47]. To become cost-effective, this community-based dengue control would need to be integrated and share resources with other public health and poverty-reduction programs. A modeled assessment of screening and vaccination in Colombia with the first licensed dengue vaccine (Sanofi's Dengvaxia) gave cost-effectiveness ratios relative to GDP per capita ranging from 0.47 (in areas with 90% dengue seropositivity among 9 year-olds) to 6.72 (with 10% dengue seropositivity) [48]. This strategy proved more cost-effective (lower ICER) as the percentage of nine-year old seropositive individuals in the population increased.

The second dengue vaccine, TAK-003 (Takeda's Qdenga), was licensed by the European Medicines Agency in 2022 and received pre-approval by the World Health Organization in 2024 [49]. Published trial results showed TAK-003 reduced dengue fever cases by 80% and, unlike Dengvaxia, created no added risk for persons with no prior dengue infection [49]. Preliminary economic models by the manufacturer projected that Qdenga would be cost saving in Puerto Rico [50] and Thailand [51]. Fig 5(B) showed that in the 9 of 11 target municipalities with dengue incidence of at least 500 per 100,000 population, *Wolbachia* also proved cost saving.

As resources for public health interventions are limited, it is informative to compare the cost-effectiveness of *Wolbachia* against that of two other public health preventive interventions in Colombia. First, a year after Colombia had introduced human papillomavirus (HPV) vaccination into its national vaccination program, [52] a cost-effectiveness analysis reported its ICER was greater than three times Colombia's then GDP per capita, so HPV was not then considered cost-effective [53]. Second, a campaign to encourage COVID-19 vaccination among those at highest risk proved cost-effective [54] by the latest criteria [43], but not cost saving.

In addition to benefit-cost and cost-effectiveness ratios, policy makers must also consider the affordability of any proposed program. The first-year costs of *Wolbachia* deployment (US$4.05 per person) represent a notable 0.8% of Colombia's 2019 per capita health expenditure and might appear too expensive if widely implemented at once. However, the program can become more affordable by phasing deployment across parts of a city over multiple years (as happened in Cali) or sequencing successive cities in different years.

If a city wished to implement a *Wolbachia* program, a portfolio of financing approaches meritconsideration. Colombia and other middle-income countries could request donor support through concessionary loans or conventional or social impact grants to advance an innovative and economically favorable approach. In domestic funding all three levels of government (municipal, department, and national) deserve attention. Colombia also has renowned private philanthropic institutions, such as the Foundation for Education and Social Development (FES), that seek to improve Colombians' health. Large employers might help fund *Wolbachia* to make their communities healthier and more attractive to workers and their families.

Several limitations should be acknowledged. The number of dengue cases differs between the RIPS and *Suficiencia* databases, pointing to inconsistencies and/or under-reporting. Second, differences in the number of dengue cases treated among different epidemiological models, macro-costing, RIPS, and SIVIGILA creates uncertainty around the estimated healthcare cost offsets. Finally, our adjustments for underreporting and misdiagnosis are based on our panel's expert

judgment rather than objective information. However, the extremely favorable benefit-cost ratios in 9 or our 11 target cities indicate that *Wolbachia* deployment would still be highly favorable in those cities.

Key strengths also deserve highlighting. First, we believe this is the first economic evaluation of *Wolbachia* in Colombia, building on the empirical record of efficacy and feasibility from the trial in Indonesia [8] and controlled observational studies in Colombia [3,13–16]. Second, we used an empirical method for valuing indirect benefits based on the overall economy [43]. As 64% of Colombia's workers were in the informal sector in 2020 and generally earned less than formal sector workers, [55] this approach is more realistic than applying formal sector wages to all cases to estimate indirect benefits, including those not employed or working informally, as was done elsewhere [20]. Third, this study's number of sites (11) substantially exceeds the numbers in previous economic analyses--3 in Indonesia [18] and 7 in Brazil [20]. These multiple sites provided the insight that not only was *Wolbachia* beneficial overall, but it was especially valuable in cities with high dengue incidence. High population density in the release area was associated with somewhat more favorable outcomes. As a square kilometer with high dengue incidence and high population density is one with substantial dengue burden, deploying *Wolbachia* in such a location will generate substantial economic value. Conversely, areas with relatively low incidence and low density would benefit much less; there another control strategy may be preferable [56].

## Conclusions

In conclusion, *Wolbachia* proved economically beneficial in all 11 target cities and cost saving (paying for itself through treatment costs averted) in the 9 target cities with adjusted incidence of at least 500 per 100,000 population. In the future, policy makers may have a portfolio of options to control dengue. *Wolbachia* is likely to be the more cost-effective or cost-saving option in municipalities with both high incidence of dengue and high population density, whereas areas with high dengue incidence but low population density should consider vaccination.

## Transparency statement

The lead author Donald S. Shepard affirms that this manuscript is an honest, accurate, and transparent account of the study being reported; that no important aspects of the study have been omitted; and that any discrepancies from the study as planned (and, if relevant, registered) have been explained.

## Supporting Information

**S1 Table. Input data for target cities.**
(PDF)

**S2 Table. Macro-costing approach to estimate the average cost of an outpatient visit and hospitalization (monetary amounts in 2020 US$).**
(PDF)

**S3 Table. Health care cost of dengue cases by type of dengue diagnosis and setting using macro-costing (amounts in 2019–2020 US$).**
(PDF)

**S4 Table. Cost of dengue case by type based on SOAT tariff schedule and macro-costing (2019–20 US$).**
(PDF)

**S1 Text. Macro-costing approach.**
(PDF)

**S2 Text. Exploratory approach of costing by health system tier.**
(PDF)

## Acknowledgments

The authors thank Luz Villarreal Salazar for serving on the study's epidemiologic panel, Ivan Velez and Patricia Arbelaez Montoya from the WMP Colombia and Reynold Dias and Katherine Anders from the global WMP (Australia) for *Wolbachia* cost data and valuable comments, reviewers for constructive comments, and Clare L. Hurley of Brandeis University for editorial assistance.

## Author contributions

**Conceptualization:** Donald S. Shepard.

**Data curation:** Samantha R. Lee, Yara A. Halasa-Rappel, Carlos Willian Rincon Perez, Arturo Harker Roa.

**Formal analysis:** Donald S. Shepard, Samantha R. Lee, Yara A. Halasa-Rappel, Carlos Willian Rincon Perez.

**Funding acquisition:** Donald S. Shepard.

**Methodology:** Donald S. Shepard, Carlos Willian Rincon Perez, Arturo Harker Roa.

**Supervision:** Donald S. Shepard, Arturo Harker Roa.

**Writing – original draft:** Samantha R. Lee, Yara A. Halasa-Rappel.

**Writing – review & editing:** Donald S. Shepard, Samantha R. Lee, Yara A. Halasa-Rappel, Carlos Willian Rincon Perez, Arturo Harker Roa.

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
