## [Decision Letter · Decision Letter 0]

14 Oct 2024

PONE-D-24-24121Economic evaluation of Wolbachia deployment in Colombia: A modeling studyPLOS ONE

Dear Dr. Shepard,

Thank you for submitting your manuscript to PLOS ONE. After careful consideration, we feel that it has merit but does not fully meet PLOS ONE’s publication criteria as it currently stands. Therefore, we invite you to submit a revised version of the manuscript that addresses the points raised during the review process.

ACADEMIC EDITOR: Please review the manuscript according to the reviewers comments

We look forward to receiving your revised manuscript.

Kind regards,

Bilal Rasool, PhD

Academic Editor

PLOS ONE

Journal Requirements:

1. When submitting your revision, we need you to address these additional requirements. Please ensure that your manuscript meets PLOS ONE's style requirements, including those for file naming. The PLOS ONE style templates can be found at https://journals.plos.org/plosone/s/file?id=wjVg/PLOSOne_formatting_sample_main_body.pdf and https://journals.plos.org/plosone/s/file?id=ba62/PLOSOne_formatting_sample_title_authors_affiliations.pdf 2. Thank you for stating the following financial disclosure: "This work was funded in whole, or in part, by the Wellcome Trust (grant 224459/Z/21/Z) to the World Mosquito Program (WMP), Monash University (Clayton, VIC, Australia).  For the purpose of open access, the author has applied a CC BY public copyright license to any Author Accepted Manuscript version arising from this submission." Please state what role the funders took in the study.  If the funders had no role, please state: "The funders had no role in study design, data collection and analysis, decision to publish, or preparation of the manuscript." If this statement is not correct you must amend it as needed. Please include this amended Role of Funder statement in your cover letter; we will change the online submission form on your behalf. 3. Thank you for stating the following in the Competing Interests section: "All authors received funding from the Wellcome Trust under a grant (224459/Z/21/Z) to the World Mosquito Program (WMP), Monash University (Clayton, VIC, Australia), which had no role in review nor the decision to submit.  The direct sponsor (WMP) had the right to review but authorized submission with no required changes.  Donald S. Shepard has received financial support from Abbott, Inc, Sanofi, and Takeda Vaccines, Inc. in the past 36 months unrelated to the present study. All other authors declare no other conflicts of interest." We note that you received funding from a commercial source: "Wellcome Trust" Please provide an amended Competing Interests Statement that explicitly states this commercial funder, along with any other relevant declarations relating to employment, consultancy, patents, products in development, marketed products, etc.  Within this Competing Interests Statement, please confirm that this does not alter your adherence to all PLOS ONE policies on sharing data and materials by including the following statement: ""This does not alter our adherence to PLOS ONE policies on sharing data and materials.” (as detailed online in our guide for authors http://journals.plos.org/plosone/s/competing-interests).  If there are restrictions on sharing of data and/or materials, please state these. Please note that we cannot proceed with consideration of your article until this information has been declared.  Please include your amended Competing Interests Statement within your cover letter. We will change the online submission form on your behalf. 4. For studies involving third-party data, we encourage authors to share any data specific to their analyses that they can legally distribute. PLOS recognizes, however, that authors may be using third-party data they do not have the rights to share. When third-party data cannot be publicly shared, authors must provide all information necessary for interested researchers to apply to gain access to the data. (https://journals.plos.org/plosone/s/data-availability#loc-acceptable-data-access-restrictions)  For any third-party data that the authors cannot legally distribute, they should include the following information in their Data Availability Statement upon submission:1) A description of the data set and the third-party source2) If applicable, verification of permission to use the data set3) Confirmation of whether the authors received any special privileges in accessing the data that other researchers would not have4) All necessary contact information others would need to apply to gain access to the data

Reviewers' comments:

Reviewer's Responses to Questions

**Comments to the Author**

1. Is the manuscript technically sound, and do the data support the conclusions?

Reviewer #1: Yes

Reviewer #2: Partly

2. Has the statistical analysis been performed appropriately and rigorously? 

Reviewer #1: I Don't Know

Reviewer #2: No

3. Have the authors made all data underlying the findings in their manuscript fully available?

Reviewer #1: Yes

Reviewer #2: No

4. Is the manuscript presented in an intelligible fashion and written in standard English?

Reviewer #1: Yes

Reviewer #2: No

5. Review Comments to the Author

Reviewer #1: The manuscript were describing an important issue about the Economic evaluation of Wolbachia deployment in Colombia, however below are some comments to the authors that may help them to improve the manuscript:

1. Line 111: All together not altogether.

2. P13: Share of Wolbacia development cost..ect, the share is from where can you please clarify more.

3. P18: Share of dengue cases correctly reported, what do you mean by that?

4. Line 130: Why the data is only available for 2019? Is it not going to affaect the accuracy of the overall results??

5. Line 134: How could the extra costs due to COVID-19 pandamic be included? on this budget evaluation which could be used for non covid pandemic periods?? I think it should be either excluded or you can compare just to let the other researchers that may use this study as a reference know that this is only applicabale if there is an oubreak or such situation.

6. Do you study all the parameters within the same time period? if not, then how do the study could be concluded.

7. Line 227: Do you mean 15 to 30 months??

8. Figure 2 needs to be adjusted and improved.

10. There are many language mistakes.

11. The whole arrangements of the paper methods and results need to be revised.

thank you very much and all the best with the revision

Reviewer #2: I have noted the following issues with the manuscript entitled "Economic evaluation of Wolbachia deployment in Colombia: A modeling study" by Shepard et al. seems interesting; however, serious issues in the present form of the draft, including analysis, should be clarified before considering this as a review.

1. It is unclear where the datasets for reported dengue cases were obtained over the years.

2. It is important to clarify if the economic analysis was conducted using anonymous datasets.

3. The interpretation of "unreported dengue cases" needs to be explained, along with the source of the datasets.

4. The unanswered comment of the previous review: “I found it unclear why (and how) mandatory road traffic tariffs being used for the cost of dengue cases. In the discussion, a macro-costing approach is mentioned but I could not see this in the methods.

5. It is crucial to address whether other mosquito control options were used in Wolbachia deployment areas over the years, and if the study is based on field or assumed data.

6. Clarify if the data is cumulative over ten years or from a single year.

7. Figure 2 is missing information due to lack of edits.

8. The methodology should explain how Wolbachia deployment occurred in the study areas over time.

9. Details on the duration and doses of Wolbachia deployment in the study areas should be provided, along with information on whether other control options were used. The map of the localities may also be added.

Additionally, the MM section is unclear, and abbreviations used in the manuscript should be written in full when first used.

Lines 165-167 states a 37.5% reduction in dengue cases overall in the first year, but it's unclear which year is the first year. The authors should adhere to the journal's instructions for writing all sections from Abstract to Conclusion, as well as for equations, formulas, references and other inaccuracies, etc.

6. PLOS authors have the option to publish the peer review history of their article (what does this mean? ). If published, this will include your full peer review and any attached files.

**Do you want your identity to be public for this peer review?** For information about this choice, including consent withdrawal, please see our Privacy Policy .

Reviewer #1: **Yes: ** Sara Abdelrahman Abuelmaali

Reviewer #2: No

---

## [Author Response · Author response to Decision Letter 0]

23 Nov 2024

Response to reviewers re PONE-D-24-24121, “Economic evaluation of Wolbachia deployment in Colombia: A modeling study”

For convenience, we have labeled comments from the editor beginning with E, those from reviewer 1 beginning with R1, and those from reviewer 2 beginning with R2. The authors’ response follows each comment. Line numbers are approximate.

E1 Please include the following items when submitting your revised manuscript:

• An unmarked version of your revised paper without tracked changes. You should upload this

E1 Authors: We attach these documents.

E2 Thank you for stating the following financial disclosure: "This work was funded in whole, or in part, by the Wellcome Trust (grant 224459/Z/21/Z) to the World Mosquito Program (WMP), Monash University (Clayton, VIC, Australia). For the purpose of open access, the author has applied a CC BY public copyright license to any Author Accepted Manuscript version arising from this submission."

E2 Authors: We address the funding statement under E3 below. We have revised role of the funder which appears in the Cover Letter. It reads as follows:

Role of the funder: The Wellcome Trust had no role in review nor the decision to submit. The direct sponsor (WMP) had the right to review but authorized submission with no required changes.

E3 Thank you for stating the following in the Competing Interests section: "All authors received funding from the Wellcome Trust, a registered charity in England Wales, under a grant (224459/Z/21/Z) to the World Mosquito Program (WMP), Monash University (Clayton, VIC, Australia, with a subaward to Brandeis University, USA, which had no role in review nor the decision to submit. The prime grantee (WMP) had the right to review but authorized submission with no required changes. Donald S. Shepard has also received financial support from Abbott, Inc, Sanofi, and Takeda Vaccines, Inc. in the past 36 months unrelated to the present study. All other authors declare no other conflicts of interest."

E3 Authors. We have revised the funding statement as shown below and placed it in the cover letter. It reads as follows:

Funding Statement: This study was funded by the Wellcome Trust, a registered charity in England Wales, under a grant (224459/Z/21/Z) to the World Mosquito Program, Monash University (Clayton, VIC, Australia) with a subaward to Brandeis University, USA. For the purpose of open access, the author has applied a CC BY public copyright license to any Author Accepted Manuscript version arising from this submission.

E4 We note that you received funding from a commercial source: Wellcome Trust" Please provide an amended Competing Interests Statement that explicitly states this commercial funder, along with any other relevant declarations relating to employment, consultancy, patents, products in development, marketed products, etc. Within this Competing Interests Statement, please confirm that this does not alter your adherence to all PLOS ONE policies on sharing data and materials by including the following statement: ""This does not alter our adherence to PLOS ONE policies on sharing data and materials.” (as detailed online in our guide for authors http://journals.plos.org/plosone/s/competing-interests [journals.plos.org]). If there are restrictions on sharing of data and/or materials, please state these. Please note that we cannot proceed with consideration of your article until this information has been declared.

E4 Authors: Our updated competing interests statement appears in the cover letter as requested and reads as follows:

Competing interests: All authors received funding from the Wellcome Trust, a registered charity in England Wales, under a grant (224459/Z/21/Z) to the World Mosquito Program (WMP), Monash University (Clayton, VIC, Australia) with a subaward to Brandeis University, USA. This does not alter our adherence to PLOS ONE policies on sharing data and materials. Donald S. Shepard has received financial support from Abbott, Inc, Sanofi, and Takeda Vaccines, Inc. in the past 36 months unrelated to the present study. All other authors declare no other competing interests.

E5 For studies involving third-party data, we encourage authors to share any data specific to their analyses that they can legally distribute. PLOS recognizes, however, that authors may be using third-party data they do not have the rights to share. When third-party data cannot be publicly shared, authors must provide all information necessary for interested researchers to apply to gain access to the data. (https://journals.plos.org/plosone/s/data-availability#loc-acceptable-data-access-restrictions [journals.plos.org])

E5 Authors: In the Data Availability Statement around line 510 the authors describe the underlying data sets and how they may be accessed.

R1.0 Reviewer #1: The manuscript were describing an important issue about the Economic evaluation of Wolbachia deployment in Colombia, however below are some comments to the authors that may help them to improve the manuscript:

R1.0 Authors. Thank you. We have responded to the specific points below.

R1.1. Line 111: All together not altogether.

R1.1 Authors. Done

R1.2 P13: Share of Wolbachia development cost..etc.., the share is from where can you please clarify more.

R1.2 Authors: We updated the explanation of the accelerated timeline (see line 285ff).

R1.3 P18: Share of dengue cases correctly reported, what do you mean by that?

R1.3 Authors. We added an explanation in lines 198ff.

R1.4 Line 130: Why the data is only available for 2019? Is it not going to affect the accuracy of the overall results??

R1.4 Authors. We have explained that more recent data, being pandemic years, would not have provided a representative basis for future projections and clarified our cost projections for COVID and non-COVID periods (see lines 143ff)

R1.5 Line 134: How could the extra costs due to COVID-19 pandemic be included? on this budget evaluation which could be used for non COVID endemic periods?? I think it should be either excluded or you can compare just to let the other researchers that may use this study as a reference know that this is only applicable if there is an outbreak or such situation.

R1.5 Authors. As explained in R1.4, we have used estimates from non-COVID periods to make projections for all cities except Cali, which would be expected to proceed in non-COVID periods.

R1.6 Do you study all the parameters within the same time period? if not, then how do the study could be concluded.

R1.6 Authors: Our cost data are generally for 2019, the last pre-pandemic year, adjusted to 2020 prices. Numbers of dengue cases, however, are annual averages over the decade ending in 2019. This averaging was done to provide a stable, representative estimate of the annual number of future cases expected with no intervention. As dengue is a communicable infectious disease, its incidence of dengue varies several fold from one year to the next, so data for the single newest year would not have provided a representative basis of future planning (see lines 179ff).

R1.7 Line 227: Do you mean 15 to 30 months??

R1.7 Authors. We rewrote lines 229 through 232 to explain our time projections being initially 30 months and subsequently revised to 15 months.

R1.8 Figure 2 needs to be adjusted and improved.

R1.8 Authors. As suggested, we replaced the former Figure 2 by an improved version. Because of the addition of the requested map, this has now been renumbered as Figure 3. We changed the title of the figure and the labels for the two lines, removed the confusing green leader lines, updated the axis labels, and put data values in the same color as their corresponding lines. To clarify further, the explanation of the new Figure 3 in the results (around lines 393ff) has also been revised.

R1.10 (Note: There was no item 9 in the list we received.) There are many language mistakes.

R1.10 Authors: We carefully reviewed and revised the manuscript for completeness, flow, clarity and accuracy.

R1.11 The whole arrangements of the paper methods and results need to be revised.

R1.11 Authors: We have revised the entire manuscript, including these two sections, by removing some inessential items, reordering sections, adding subheadings, and editing the text.

R1.12 Thank you very much and all the best with the revision

R1.12 Authors: We thank the reviewer for the many constructive suggestions.

R2.0 Reviewer #2: I have noted the following issues with the manuscript entitled "Economic evaluation of Wolbachia deployment in Colombia: A modeling study" by Shepard et al. seems interesting; however, serious issues in the present form of the draft, including analysis, should be clarified before considering this as a review.

R2.0 Authors: As noted in R1.10 and R1.11, we carefully reviewed and revised the manuscript for completeness, flow, clarity and accuracy.

R2.1 It is unclear where the datasets for reported dengue cases were obtained over the years.

R2.1 Authors: At the start of the section “Disease burden of dengue,” we added a paragraph explaining how the data were obtained from SIVIGILA (lines 187ff).

R2.2 It is important to clarify if the economic analysis was conducted using anonymous datasets.

R2.2 Authors. All economic analyses were strictly anonymous. We added an explanatory paragraph (lines 245ff).

R2.3 The interpretation of "unreported dengue cases" needs to be explained, along with the source of the datasets.

R2.3 Authors. We expanded the discussion of SIVIGILA and unreported cases in the section “Disease burden of dengue” (lines 176ff).

R2.4 The unanswered comment of the previous review: “I found it unclear why (and how) mandatory road traffic tariffs being used for the cost of dengue cases. In the discussion, a macro-costing approach is mentioned but I could not see this in the methods.

2.4 Authors. We explained how the SOAT tariffs serve as a benchmark for all insurers (lines 219ff) and added two paragraphs to clarify macro-costing and its potential (lines226-249ff).

R2.5 It is crucial to address whether other mosquito control options were used in Wolbachia deployment areas over the years, and if the study is based on field or assumed data.

R2,5 Authors. In Yogyakarta, there was some focal insecticide spraying in response to certain notified dengue cases. However, rather than confounding the estimated benefit of Wolbachia, it actually offset some of the benefits. If there was any impact, it made the estimated benefit of Wolbachia conservative. For an explanation, see the discussion of parameters P4 through P8 (lines 155ff).

R2.6 Clarify if the data is cumulative over ten years or from a single year.

R2.6 Authors. We have edited Tables 2 and 3 to ensure consistent terminology and present the data in the form needed for the analysis. Most items are cumulative present value over 10 years. The labels all columns with cumulative values begin by the abbreviation PV. For dengue cases, however, we present average annual cases in the release area of each target city.

R2.7 Figure 2 is missing information due to lack of edits.

R2.7 Authors. As described in R1.8 we have created a new version of Figure 2.

R2.8 The methodology should explain how Wolbachia deployment occurred in the study areas over time.

R2.8 Authors. We describe the implementation process in Cali, which serves as a blueprint for the other target cities, under “Cost of Wolbachia deployment” (lines 272-293).

R2.9 Details on the duration and doses of Wolbachia deployment in the study areas should be provided, along with information on whether other control options were used. The map of the localities may also be added.

R2.9 Authors. We describe the original plans and indicate how experience allowed the process to be accelerated, reducing the projected duration from 30 to 15 months per target city. See R2.8 and the section “Cost of Wolbachia deployment” (lines 274ff). A map of Colombia showing the target areas was created (see Figure 1). We renumbered the other four figures.

R2.10 Additionally, the MM section is unclear, and abbreviations used in the manuscript should be written in full when first used.

R2.10 Authors. We have reviewed the materials and methods, as well as other parts of the methods, correcting all errors we found.

R2.11 Lines 165-167 states a 37.5% reduction in dengue cases overall in the first year, but it's unclear which year is the first year. The authors should adhere to the journal's instructions for writing all sections from Abstract to Conclusion, as well as for equations, formulas, references and other inaccuracies, etc.

R2.11 Authors. In the section “Disease burden of dengue,” we have explained the time schedule so that planning occurs in year 0 (the decision year), and deployment or release begins at the start of year 1 (see lines 207ff).

E6 Do you want your identity to be public for this peer review? For information about this choice, including consent withdrawal, please see our Privacy Policy [c05y1x9s.r.us-east-2.awstrack.me].

Reviewer #1: Yes: Sara Abdelrahman Abuelmaali

Reviewer #2: No

E6 Authors. We appreciate the comments and suggestions from the editor and both reviewers. We would like to thank Sara Abdelrahman Abuelmaali and an anonymous reviewer. We added a phrase in the acknowledgments to thank the reviewers.

E7 In a subsequent communication (Nov 14) the editor asked us to ensure the content and placement of our data availability statement conformed to the journal's policies.

E7 Authors. We removed the data availability statement from the manuscript. We created a new data availability statement, based in part on that information, and included it in the additional information.

---

## [Decision Letter · Decision Letter 1]

29 Jan 2025

PONE-D-24-24121R1Economic evaluation of Wolbachia deployment in Colombia: A modeling studyPLOS ONE

Dear Dr. Shepard,

Thank you for submitting your manuscript to PLOS ONE. After careful consideration, we feel that it has merit but does not fully meet PLOS ONE’s publication criteria as it currently stands. Therefore, we invite you to submit a revised version of the manuscript that addresses the points raised during the review process.

**ACADEMIC EDITOR: Please revise the manuscript according to the reviewers comments**

We look forward to receiving your revised manuscript.

Kind regards,

Bilal Rasool, PhD

Academic Editor

PLOS ONE

Reviewers' comments:

Reviewer's Responses to Questions

**Comments to the Author**

1. If the authors have adequately addressed your comments raised in a previous round of review and you feel that this manuscript is now acceptable for publication, you may indicate that here to bypass the “Comments to the Author” section, enter your conflict of interest statement in the “Confidential to Editor” section, and submit your "Accept" recommendation.

Reviewer #1: All comments have been addressed

Reviewer #3: (No Response)

2. Is the manuscript technically sound, and do the data support the conclusions?

Reviewer #1: Yes

Reviewer #3: Yes

3. Has the statistical analysis been performed appropriately and rigorously? 

Reviewer #1: Yes

Reviewer #3: Yes

4. Have the authors made all data underlying the findings in their manuscript fully available?

Reviewer #1: Yes

Reviewer #3: Yes

5. Is the manuscript presented in an intelligible fashion and written in standard English?

Reviewer #1: Yes

Reviewer #3: Yes

6. Review Comments to the Author

Reviewer #1: I am reviewing this article for the second time, and I would like to commend the authors on producing a good and impactful scientific piece. All previous comments have been addressed adequately. However, I have a few additional comments:

Line 147, Page 11: Please clarify whether "P!" should read "P1" or if it is indeed a typographical error (P! vs. P1).

Lines 194-196, Page 13: I believe this paragraph would be more appropriately placed in the discussion section rather than the results.

Thank you for considering these points.

Reviewer #3: "Economic evaluation of Wolbachia deployment in Colombia: A modeling study" by Shepard et al. appears to be a valuable contribution; however, the authors need to improve several key aspects to enhance the manuscript's suitability for publication.

Lines 59-61: Please include the number of DENV cases during the specified years and the associated costs. This information is fundamental from an economic perspective.

Lines 72-74: It is crucial to clarify which Wolbachia strain was used in Australia and to describe the environmental conditions of Cairns. Additionally, clarify whether the same Wolbachia strain will be used in Colombia.

General: Ensure that all scientific names, such as Wolbachia, are italicized consistently throughout the manuscript.

Lines 140-141: Please specify the source of this information.

Lines 142-144: The authors should consider that the CRT conducted in Yogyakarta has different socio-economic implications when extrapolated to Colombia. This factor must be adjusted based on the analysis.

Line 160:"P1" should be corrected (P!).

Line 229: The statement that the Wolbachia program in Colombia will result in a 75% reduction in dengue cases appears to be based on Yogyakarta's results. If this percentage is used as a fixed estimate, it could pose a risk in the coming years. It is recommended to present a range instead of a fixed percentage.

Lines 309-312: Among the listed cost categories, which are the most expensive? The authors should propose or estimate strategies to reduce costs, potentially through intersectoral funding, including contributions from private companies.

Table 2: What is the budget allocated by each city for vector control? This information is crucial for determining the feasibility of the PR implementation.

Discussion

The manuscript should discuss how each city (municipality) will formalize agreements and incorporate the PR strategy as part of the national dengue control strategy. Additionally, consider how private companies could contribute to these efforts.

Lines 532-534: It is important to incorporate the most common risks associated with costs into the model. Other potential risks, such as pandemics, changes in government, and budget reductions, should also be considered.

Lines 568-574: The authors could propose a hybrid model combining vaccines and traditional vector control measures to reduce implementation costs. This approach could facilitate cost-sharing with other institutions and enhance the accessibility and affordability of the PR strategy.

Conclusion

Lines 610-613: Municipalities generally have very limited budgets (especially in LATAM), and mobilizing resources requires scaling up efforts to the national level. A gradual integration of the Wolbachia strategy within a holistic integrated vector control framework is necessary. Otherwise, a standalone strategy may be financially unfeasible.

7. PLOS authors have the option to publish the peer review history of their article (what does this mean? ). If published, this will include your full peer review and any attached files.

**Do you want your identity to be public for this peer review?** For information about this choice, including consent withdrawal, please see our Privacy Policy .

Reviewer #1: **Yes: ** Sara Abdelrahman Abuelmaali

Reviewer #3: No

---

## [Author Response · Author response to Decision Letter 1]

11 Mar 2025

Response to reviewer

We thank the reviewer for the constructive comments. In the course of responding to these comments, we also incorporated some editorial refinements.

Reviewer 3, Lines 59-61: Please include the number of DENV cases during the specified years and the associated costs. This information is fundamental from an economic perspective.

Response: We have expanded and renumbered the tables to incorporate this. The new Table 2 has two estimates of the number of dengue cases. In column 4 we report the average notified release area dengue cases. In column 5 we report the projected release area dengue cases without treatment. The latter number includes adjustments for under-reporting. In column 3 of the new Table 3 we report the current cost of dengue cases (without the Wolbachia intervention).

Reviewer 3, Lines 72-74: It is crucial to clarify which Wolbachia strain was used in Australia and to describe the environmental conditions of Cairns. Additionally, clarify whether the same Wolbachia strain will be used in Colombia.

Response. We expanded the phrase around Cairns, Australia, adding “using Ae. aegypti infected with the wMel strain of Wolbachia" (see line 67). Under the replacement strategy, we revised the sentence to read ” Singapore releases only male Wolbachia infected mosquitoes (wAlbB strain)." (see line 72). We modified the sentence about the Yogyakarta to read “found that the Wolbachia (wMel strain) replacement strategy reduced all virologically-confirmed symptomatic dengue cases…” (line 79). We modified the sentence about Niteroi, Brazil to read “wMel-Wolbachia reduced the incidence of dengue by 69%,..." (line 87). We modified the background on Colombia to read "In Colombia, pilot wMel-Wolbachia releases began in the city of Bello..." (line 92).

Reviewer 3, General: Ensure that all scientific names, such as Wolbachia, are italicized consistently throughout the manuscript.

Response. We have checked the manuscript and capitalized as necessary.

Reviewer 3, Lines 140-141: Please specify the source of this information.

Response. We modified Table 1 to change the column heading to “Description and source” and added the source for each item.

Reviewer 3, Lines 142-144: The authors should consider that the CRT conducted in Yogyakarta has different socio-economic implications when extrapolated to Colombia. This factor must be adjusted based on the analysis.

Response. We addressed this concern in the discussion in the paragraph beginning “As a Wolbachia program depends….” (line 517)

Reviewer 3, Line 160:"P1" should be corrected (P!).

Response. Done

Reviewer 3, Line 229: The statement that the Wolbachia program in Colombia will result in a 75% reduction in dengue cases appears to be based on Yogyakarta's results. If this percentage is used as a fixed estimate, it could pose a risk in the coming years. It is recommended to present a range instead of a fixed percentage.

Response. To address this important question, we added sentences in the discussion beginning “Our projected 75% efficacy is likely conservative…” (line 532)

Reviewer 3, Lines 309-312: Among the listed cost categories, which are the most expensive? The authors should propose or estimate strategies to reduce costs, potentially through intersectoral funding, including contributions from private companies.

Response. We added the sentences beginning “In Cali, the most expensive…” (line 488)

Reviewer Table 2: What is the budget allocated by each city for vector control? This information is crucial for determining the feasibility of the PR implementation.

Response: Table 3, Column 4 shows this current cost. We added a comment to the discussion beginning with “While we project reductions in conventional vector control….” (line 485)

Reviewer 3, Discussion. The manuscript should discuss how each city (municipality) will formalize agreements and incorporate the PR strategy as part of the national dengue control strategy. Additionally, consider how private companies could contribute to these efforts.

Response. We added the sentences beginning: “If a city wished to implement a Wolbachia program, a portfolio of financing approaches….” (line 567)

Reviewer 3, Lines 532-534: It is important to incorporate the most common risks associated with costs into the model. Other potential risks, such as pandemics, changes in government, and budget reductions, should also be considered.

Response: We added a paragraph to discuss risks and resilience of the Wolbachia approach beginning “Any public health program risks interference….” (line 514).

Reviewer 3, Lines 568-574: The authors could propose a hybrid model combining vaccines and traditional vector control measures to reduce implementation costs. This approach could facilitate cost-sharing with other institutions and enhance the accessibility and affordability of the PR strategy.

Response: We responded insofar as our data allowed by revising our conclusion with the sentence beginning “Wolbachia is likely to be the more cost-effective or cost-saving …” (line 605). More specific recommendations would require the future development and calibration of more detailed epidemiologic and economic models.

Reviewer 3, Conclusion, Lines 610-613: Municipalities generally have very limited budgets (especially in LATAM), and mobilizing resources requires scaling up efforts to the national level. A gradual integration of the Wolbachia strategy within a holistic integrated vector control framework is necessary. Otherwise, a standalone strategy may be financially unfeasible.

Response: We added a sentence to the discussion beginning “To become cost-effective, this community-based dengue control…” (line 540) We believe that previously mentioned revisions also address this comment. See the sentences on financing “…a portfolio of financing approaches deserve consideration….” (lines 567) and the revision to the conclusions “Wolbachia is likely to be the more cost-effective or cost-saving …” (line 605)

---

## [Editor Report · Decision Letter 2]

18 Mar 2025

Economic evaluation of Wolbachia deployment in Colombia: A modeling study

PONE-D-24-24121R2

Dear Dr. Shepard,

We’re pleased to inform you that your manuscript has been judged scientifically suitable for publication and will be formally accepted for publication once it meets all outstanding technical requirements.

Kind regards,

Bilal Rasool, PhD

Academic Editor

PLOS ONE

Additional Editor Comments (optional): The Journals'  authors instructions related formatting style, and technical requirements may be addressed.

---

## [Editor Report · Acceptance letter]

PONE-D-24-24121R2

PLOS ONE

Dear Dr. Shepard,

I'm pleased to inform you that your manuscript has been deemed suitable for publication in PLOS ONE. Congratulations! Your manuscript is now being handed over to our production team.

Kind regards,

on behalf of

Dr Bilal Rasool

Academic Editor

PLOS ONE